# Impact of Motivational Workshop on Financial Inclusion of Rural People in Bangladesh: Evidence from Randomized Controlled Trial

Md Monzur Morshed [ID] and Keshav Lall Maharjan *[ID]

International Economic Development Program, Graduate School of Humanities and Social Sciences, Hiroshima University, Higashi-Hiroshima 739-8529, Japan; monzur_engdu@yahoo.com
* Correspondence: mkeshav@hiroshima-u.ac.jp

**Abstract:** Despite the expansion of financial institutions and the proliferation of mobile financial services, reaching the unbanked and bringing them under formal financial services has become a policy concern in many developing countries. Due to the lack of financial accounts, unbanked people prefer informal, risky, and inconvenient mechanisms for receiving, sending, and transferring money. Previous studies rely much on common interventions like no account maintenance and opening fees, easy documentation processes, and money subsidies for opening financial accounts. This study aims to examine the impact of the motivational workshop on opening savings accounts through causality among the unbanked people in a setting where the respondents are unbanked despite having all the requirements and many institutional offers to open savings accounts. We encouraged the unbanked people through a one-hour-long motivational workshop to open savings accounts. Based on our cross-sectional data and randomized controlled trial experiment among the 505 unbanked rural people at Dhubil union under Sirajganj in Bangladesh, we have evidence that motivational workshop positively impacts opening accounts by 32.33 percent. However, the account opening rate differs in terms of respondent's preference for financial institutions. Our study also finds that unbanked people have the highest preference for mobile financial services for opening accounts resulting in 15.33 percent. The result of this study has some policy implications for adopting effective strategies for universal financial access in many developing countries.

**Keywords:** financial inclusion; financial access; unbanked; motivational workshop; randomized controlled trial

## 1. Introduction

Financial exclusion is a barrier against mainstreaming the unbaked in formal financial system. On the contrary, Abdul Razak and Asutay (2022) termed financial inclusion as an instrument for economic well-being, moreover financial inclusion is a tool of achieving sustainable development goals (Kara et al. 2021), and women empowerment (Siddik 2017). Although financial inclusion positively impacts on reducing rural and urban income gap (Sun and Tu 2023), and income inequality (Valdebenito and Pino 2022), only 53% of Bangladeshi adults have bank accounts (Demirgüç-Kunt et al. 2022). Even though, the number of unbanked people is decreasing globally still, a remarkable gap is observed from developed to developing countries, region to region, urban to rural, and even between males and females regarding access to financial institutions (Demirgüç-Kunt et al. 2022; Dupas et al. 2018). According to the Global Findex Report 2021, globally 1.4 billion people are financially excluded. Simply they do not have an account in financial institutions. Demirgüç-Kunt et al. (2020); Allen et al. (2016) find that typically, in every context poor adult, marginalized people, ethnic minorities, and women are generally financially excluded. Demirguc-Kunt and Klapper (2012); Allen et al. (2016) provide a pertinent example of the diversified picture is that account ownership in developed countries is

almost 91%, whereas it is only 41% in developing countries. Although bank accounts are an important part of daily economic activities, developing countries are still lagging in achieving universal financial access (Dupas et al. 2018; Asuming et al. 2019).

Due to informal livelihood strategies (Conroy 2005), The unbanked may have less income but they have cash flow for survival. Because of informal and irregular income sources, they do not consider financial access. These uncertainty and vulnerability increase the propensity of being defaulter (Mogaji et al. 2021). The lack of access to financial services negatively impacts on social inclusion as financial exclusion is synonym of social exclusion (Sarma and Pais 2011). In contrast, Demirgüç-Kunt et al. (2022) further assert that financially included people have the opportunity to mitigate unexpected natural shock, income loss or crop loss since they have financial accounts for borrowing money and getting government benefits quickly. Due to the lack of financial accounts, the unbanked people have to rely on risky, expensive and inconvenient ways of receiving, sending and transferring money (Dupas et al. 2018). Being unbanked, they store money under pillow, mattress or keep cash in hand which is prone to expenditure. For the lack of financial access, the unbanked people are exploited through the high interest rate charged by the local money-lenders (Mogaji et al. 2021).

Many studies deal with the unbanked people who lack identity documents (Chin et al. 2010), physical proximity to financial institutions (Prina 2015). Moreover, zero account opening and maintenance fees, money subsidy, easing application process, financial education (Grimes et al. 2010; Schoofs 2022; Yan and Qi 2021), alone with financial training (Dupas et al. 2018; Kostov et al. 2014; Hoy et al. 2022; Cole et al. 2011), have already been experimented in many developing countries. No experimental studies have been found on the unbanked based on sharing information related to financial access and inspiring for opening accounts which may instigate their financial inclusivity. Consequently, this study shades light on those unbanked section of people who have all requirements and documents to initiate a financial account. Self-exclusion or lack of knowledge may be the prime reasons for being unbanked in our context. Moreover, this study further goes one step ahead of loan accounts for example the ownership of a savings account so that the unbanked may have permanent transactional facility for financial transaction. Thus, this study has several aims. Firstly, the study aims to increase opening savings accounts by the unbanked in a setting where the government has adopted policies for sustainable financial inclusion strategies by offering "No-Frill" accounts (accounts having no opening and yearly maintenance charges). The second aim is to observe the response of poor, elderly people and rural marginalized people's response through causality. In this regard, we apply randomized controlled trial experiment to measure the unbiased estimation of attending a motivational workshop on opening savings accounts. We have two reasons for adopting motivational workshop as our treatment. Firstly, we consider the context of the study. The underpinning assumption is that motivation is the driving force for pursuing a positive change. Secondly, we need a cost effective and replicable intervention which will be appropriate for developing countries since low-cost intervention is suitable and viable for the policy makers (Vu et al. 2020).

In the past government distributed public transfers through a coupon or check, but with the advancement of technology, money transfers have become digitalized. Thus, public transfer is now credited to the beneficiary's accounts or cards (Fitzpatrick 2015). Therefore, policymakers have concerns with financial inclusion for developing countries since 2005 (Nur et al. 2014). Literature suggest that greater financial access is closely related to low account opening and maintaining cost, proximity of financial institutions, legal rights and political stability (Allen et al. 2016). The trend of financial inclusivity accelerates after the World Bank urges to achieve Universal Financial Access by 2020 (UFA 2020) to facilitate financial accounts or technological instrument for storing, transferring and receiving money. Nevertheless, Demirgüç-Kunt et al. (2022) assert that 1.4 billion people are unbanked worldwide. The Global Findex Report 2021 identifies Bangladesh as one of the seven countries where globally 4% unbanked people live. On the contrary, the

collaborative effort of the ministry of finance division and the central bank of Bangladesh has been manifested through the National Financial Inclusion Strategy of Bangladesh (NFIS-B) which sets a goal to achieve 100% of adults must have a financial account by 2025. The NFIS-B aims to improve inclusive and sustainable financial inclusion for all through digitalization and innovation. Thus, there is much scope of improving financial access in Bangladesh through opening a savings account by the unbanked.

Emphasizing on the financial inclusion, Sen and De (2018) assert that one of the essential steps of financial inclusion is possessing a bank account. In addition, Maity and Sahu (2022) mention that account ownership is the fundamental condition for financial inclusion. Moreover, Allen et al. (2016) adopted "opening bank accounts" as a proxy variable for measuring financial inclusion. Therefore, account ownership is the initial step of reaching financial inclusion from exclusion. Furthermore, people can transfer and transact money safely through a transactional account. In addition, emphasizing the importance of bank accounts or accounts in other financial institutions, Aguila et al. (2016) mention that marginalized and minority groups can contribute to the development of society and economy by participating in the financial system. In this regard, Iqbal et al. (2020) opine that everybody must have a bank account to transact and receive payments quickly. Bank account ownership is the entry point of the formal financial system resulting significant impact on shifting from informal to formal financial system (Belayeth Hussain et al. 2019). Financial inclusion can accelerate sustainable development and poverty alleviation. Maity and Sahu (2022) posit that the low-income groups can contribute to their social and economic upliftment by ensuring access to financial institutions. Furthermore, the National Financial Inclusion Strategy of Bangladesh (NFIS-B) views financial inclusion as an effective way for poverty reduction. Account ownership is necessary for accessing financial services such as payments, savings, insurance, debit and credit (Corrado 2020). Moreover, Maity and Sahu (2022) state that a formal financial account can encourage savings, access to credit, easy transfer of wages, remittances and government payments. In addition, inclusive access to financial institutions improves productive consumption and production patterns (Conroy 2005).

## 2. Bangladesh Bank's Initiatives for Financial Access and Related Literature

### 2.1. Bangladesh Bank's Initiatives for Financial Access

The central bank of Bangladesh, Bangladesh Bank (BB), has adopted measures for achieving inclusive and sustainable financial access. To achieve the objective, BB introduced No-Frill accounts for the farmers, the impoverished, the social safety net beneficiaries, ready-made garments workers, marginalized and vulnerable sections of society. No-frill account requires only 10 Bangladeshi Taka (currency of Bangladesh) equivalent to 0.094 USD to open an account as an initial amount. This No-frill account has no yearly maintenance charge. BB also extended free account opening options for the street urchins and working children through the assistance of NGO personnel for maintaining the account. School banking accounts are another initiative of Bangladesh Bank for the students to achieve financial inclusion. Besides, the innovative account opening provisions, BB has also launched a financial literacy program for imparting financial literacy knowledge through video clips on its website.

Bangladesh has a diversified banking system, including state-owned commercial banks, Islamic banks, specialized banks for sectoral development, and private and foreign commercial banks. According to the Bangladesh Bank's quarterly report on financial institutions 2023, currently 61 scheduled banks are operating their services through 10,942 branches in Bangladesh. BB has issued official directives to open 50% of branches in rural areas since 2010 so that the rural people can avail themselves on banking facilities. Moreover, BB has also instructed the other banks since 2013 to operate agent banking services where branch banking is not possible due to geographical barriers. Considering the proliferation of mobile phone access and technological advancement, BB has licensed mobile financial services since 2011 to many banks. Furthermore, Automated Teller Machine

(ATM) booths, Small and Medium Enterprise (SME) branches, and agricultural branches are operating in Bangladesh for smoothening banking facilities. Re-financing agricultural loans, SMEs, and women entrepreneurs is the other strategy of BB for achieving universal access to financial institutions. Besides the banking system, other financial institutions such as cooperatives, micro-financial institutions, and insurance companies are available both in urban and rural areas (Bank 2019). Despite having many financial institutions in Bangladesh, bank account ownership is low in Bangladesh. As a result, financial inclusion portrays a diversified image and exclusion rate is considerably higher in rural areas than urban areas.

*2.2. Literature Review on Financial Access*

Yan and Qi (2021) assessed the role of family education on access to financial institutions based on 22,242 data from 27 countries; their findings claim that family education positively impacts opening bank accounts by 1.9 percent. In another study, Grimes et al. (2010) examine the long-term effect of studying Economics and Business Studies on having bank accounts. They observe that high school courses in Economics and Business Studies significantly impacts on possessing a bank account. Cole et al. (2011) show a 12.70% take-up rate by concluding that financial education (knowledge) modestly affects opening accounts, whereas little subsidy (price) increases the demand for opening financial accounts. Furthermore, Blanco et al. (2020) Studied the effects of educational programs on adapting retirement accounts, my retirement accounts (myRA), among the 142 Hispanics. The study finds that 12 percent rise in opening myRA accounts among Hispanics.

In addition, Karakara and Osabuohien (2019) observed the effect of household's information and communication technology access (such as radio, television, internet, mobile phone, and computer) on bank patronage in West Africa resulting the bank patronage increases in Ghana (15%) and Burkina Faso (48%). Moreover, the inclination to ICT (Information and Communication Technology) has a positive impact on access and use of banking service (Efobi et al. 2014).

On the other hand, financial literacy training yields a positive influence on access to financial institutions, but the literature lacks concrete conclusion about the duration of training for instance a two-hour session for migrants (Gibson et al. 2014), a two-day long training varies from 4 h to 9 h for migrants (Doi et al. 2014), a two-hour session for elderly people (Bucciol et al. 2021).

Some experimental studies have been conducted on financial access and savings among unbanked people. Dupas et al. (2018) executed a randomized controlled trial experiment by randomly assigning no account opening and maintenance fee, easy form-filling process, and covering expenses for documentation through the voucher as an intervention in Malawi, Uganda, and Chile. The study reveals that the intervention has a notable impact on the initiation of opening accounts among the treatment groups in Malawi (69%), Uganda (54%), and Chile (17%). Prina (2015) also offers no opening and maintenance fee and physical proximity through the local bank branches to the slum dwellers women in Nepal. The study finds a high take-up rate (84%) and active account use during the study period. Fitzpatrick (2015) also mentions account ownership increases by 12 percent points for families with children after transferring child benefits through electronic media in the UK. Hoy et al. (2022) carry out an experimental study in Papua New Guinea and observe that the soft intervention program such as literacy workshops, no-fee packages and smoothing documentation has a positive impact on opening accounts.

The Muslim community in many countries refrained from the formal banking sector because of the lack of Islamic financial institutions. In addition, "interest" is strictly prohibited in Islam due to *Shari'ah* compliance. According to The Global Findex report 2021, due to religious concerns 10% of people worldwide are financially excluded. In this regard, Sharia-compliant credit from formal institutions can enhance the financial access of small-scale Muslim farmers in Afghanistan (Moahid and Maharjan 2020). Several studies advocate for different approaches to the financial access of Muslims such as the

Islamic finance system (Rosli et al. 2016), Islamic microfinance (Hassan 2015) and Islamic pawnbroking (Abdul Razak and Asutay 2022).

A good number of studies focus on the determinants of financial inclusion. Lotto (2018) highlights that gender, education, age, and income are vital factors for financial inclusion. Njanike (2019) identifies demographic factors such as age, gender, marital status, and education are crucial factors for getting access to financial institutions. (Hinson et al. (2009) mention that distance, accessibility, and friends' recommendations are essential factors for opening bank accounts in Ghana.

Through mobile phone technology, financial inclusion brought a remarkable change in Sub-Saharan countries. Akhter and Khalily (2020) find that households having migrants are 43% more likely to adopt mobile financial services; on the other hand, female households are less likely to take up MFS. Although digital financial services significantly bloomed during the COVID-19 situation, still poor, rural residents, and older people are lagging in adopting digital financial services (Tay et al. 2022).

### 2.3. Literature Review on Motivation

Seshan and Yang (2014) studied the impact of a "motivational workshop" on financial decision-making among Indian migrants in Qatar and observed that a motivational workshop has beneficial influences on financial decision-making. Moreover, Allen et al. (2016) mention that the characteristics of the target people need to be considered for adopting a financial inclusion policy. Phung et al. (2023) emphasized the role of motivation in intervention programs and financial literacy training. Diskin and Hodgins (2009) find a motivational session has a positive impact on reducing gambling behavior and less expenditure on gambling among gamblers. Furthermore, Mahadeva (2008) suggests that motivational training is one of the alternatives for the financial inclusion of underserved populations in India. For this, we are convinced to accept a motivational workshop as our treatment.

### 2.4. Village Defense Party

We have chosen our sample with the help of the Village Defense Party (VDP) members. The VDP is the village-level unit working under Ansar and the VDP organization. As per the organogram of the Bangladesh Ansar and VDP, every village has 64 VDP members consisting of 32 males and 32 females. According to the VDP embodiment guidelines, one must have some criteria for being a VDP member. The mandatory requirements for being a VDP are at least 18 years of age, possession of a national identity card (NID) or chairman certificate for supporting citizenship, no criminal records at police station, at least eight years of school education, and completion of a ten-day village-based basic training course. The VDP members do not receive any fixed and regular salary from the organization. They get only some remunerations based on their part-time engagements in election duty, festival duty for controlling crowds, maintaining security, checking guards, and maintaining queues for easy movement of people. VDP membership is an identity and organizational attachment, not an occupation. According to the Village Defense Party Act 1995, the VDP members provide quick responses to assist in disaster management and help law and order enforcement agencies in crisis. The VDP members are the local community people within the area. The VDP members are generally farmers, housewives, students, and employed and unemployed youth. Since the VDP members are economically marginalized in rural areas, other motivations for being a VDP member are the opportunities for free vocational training provided by the organization around the year and a 5% job quota in low-ranking third and fourth-class government jobs such as office assistance, night-guard, driver, computer operator, gardener, cook, and cleaner.

## 3. Materials and Methods

### 3.1. The Study Area

The study was conducted at Dhubil Union under the Raiganj sub-district in Sirajganj, Bangladesh. The area lies in the northern part of Bangladesh (Figure 1). In Bangladesh, the union is the smallest administrative division and the initial step of the local government system. Administratively, Dhubil Union has 17 villages. We purposively selected the study area considering the availability of financially excluded people since the district-wise financial inclusion index value of Sirajganj was 0.548 in 2014 (Hasan and Islam 2016). Most of the people depend on agriculture-based activities for their livelihood. There is no geographical barrier for road transport communication to the local *hat* (periodical open market) and *bazaar* (daily market). The area of the union is 16.75 km. There are nine government and private banks in the study area. Furthermore, many microfinance institutions, cooperatives, and mobile financing service points are available in the area.

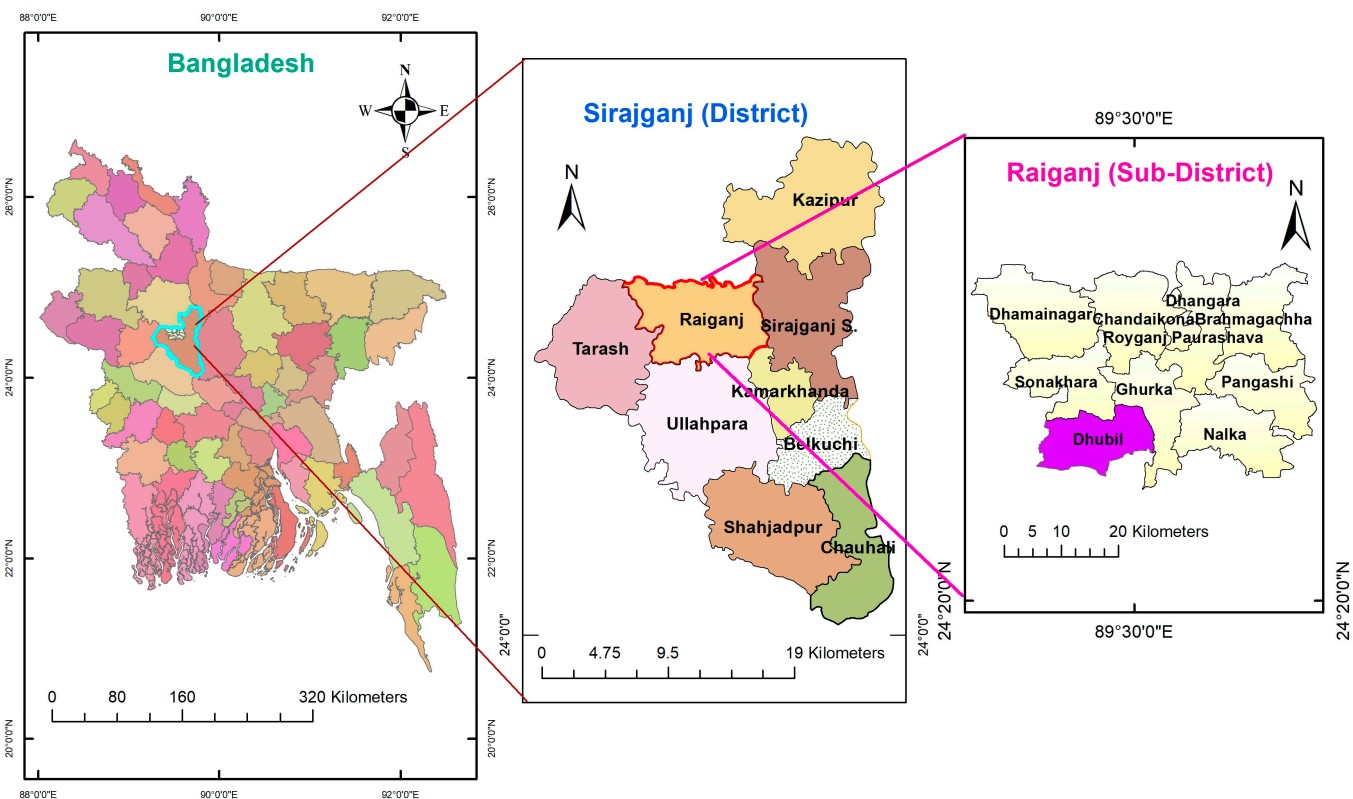

**Figure 1.** Map of the study area. Source: Authors.

### 3.2. Sampling and Data Collection

A list of the unbanked villagers was prepared with the help of the District Commandant's office of Ansar and VDP, Sirajganj since VDP is the grass root level unit in Bnagladesh. Finally, 505 villagers were identified as unbanked. For preparing the list of the unbanked, we asked whether they have any financial accounts or not. Several other questions were asked for reconfirmation of their financial exclusion such as whether he or she applied for any loan from any financial institutions last month. We even asked the aged people whether they had been nominated for any social safety net program from the government last month or not. We were conscious of creating the list to avoid the mixture of treatment and control from the same household. We did not allow two people in the sample frame from the same household. After confirming those issues, the list was prepared and finalized. The details of the village-wise sample are presented in Appendix A (Figure A1).

The baseline survey was conducted among the unbanked with the help of enumerators. We hired ten enumerators from different villages to collect data. Some enumerators had

to collect data from more than one village. A short training session was arranged for the enumerators before baseline data collection in September 2022. The baseline data was collected by visiting the respondent's residence. Some houses were revisited due to the unavailability of respondents at home since no prior information was given for visiting their houses. We have collected both demographic and household data such as age, gender, education, marital status, family size, agricultural land size, total income and total expenditure, distance from banks and other financial institutions, and possessions of mobile and internet users. Descriptions of all variables are presented in Table 1.

**Table 1.** Description of variables.

| No | Variable | Description and Measurement |
|----|----------|------------------------------|
| 1. | Age | Age of the respondent by year |
| 2. | Gender | Gender of the respondent, 1 equals male, 0 if otherwise |
| 3. | Education | Highest educational attainment by year |
| 4. | Family size | Total family members, by number |
| 5. | Agricultural land size | Agricultural land ownership of the household, by decimal |
| 6. | Total income | Respondent's monthly total income in Bangladeshi currency (BDT*) |
| 7. | Total expenditure | Respondent's monthly total expenditure by BDT |
| 8. | Mobile user | Status of mobile subscription, 1equals mobile user, 0 if otherwise |
| 9. | Internet user | Status of internet user, 1equals internet user, 0 if otherwise |
| 10. | Bank Distance | Distance from bank to respondent's house (in meters) |
| 11. | MFIs Distance | Distance from micro financial institution to respondent's house (in meters) |
| 12. | MFS distance | Distance from mobile financial service points to respondent's house (in meters) |
| | Treatment variable | |
| 13. | | Motivational workshop, 1 equals treatment group, 0 if otherwise |
| | Outcome Variable | |
| 14. | Account opening | 1 if the respondent opens an account, 0 if otherwise |
| 15. | Bank account opening | 1 if the respondent opens an account in the bank, 0 if otherwise |
| 16. | Micro financial account (MFIs)opening | 1 if the respondent opens an account in MFIs, 0 if otherwise |
| 17. | Mobile financial service (MFS) account opening | 1 if the respondent opens an account in MFS, 0 if otherwise |

Note: BDT* stands for Bangladeshi currency known as Bangladeshi Taka. 1 USD is equivalent to 110.50 BDT as of 16 October 2023 as per the Bangladesh Bank rate.

After completing baseline data, the respondents were divided into two groups, treatment and control, by simple randomization. The sample size in the treatment and the control group is 240 and 265, respectively. Figure 2 depicts research design of the study step by step.

We conducted our follow-up survey in November 2022. Finally, we successfully collected all the data since the enumerators were chosen from each village who are acquainted with the respondents.

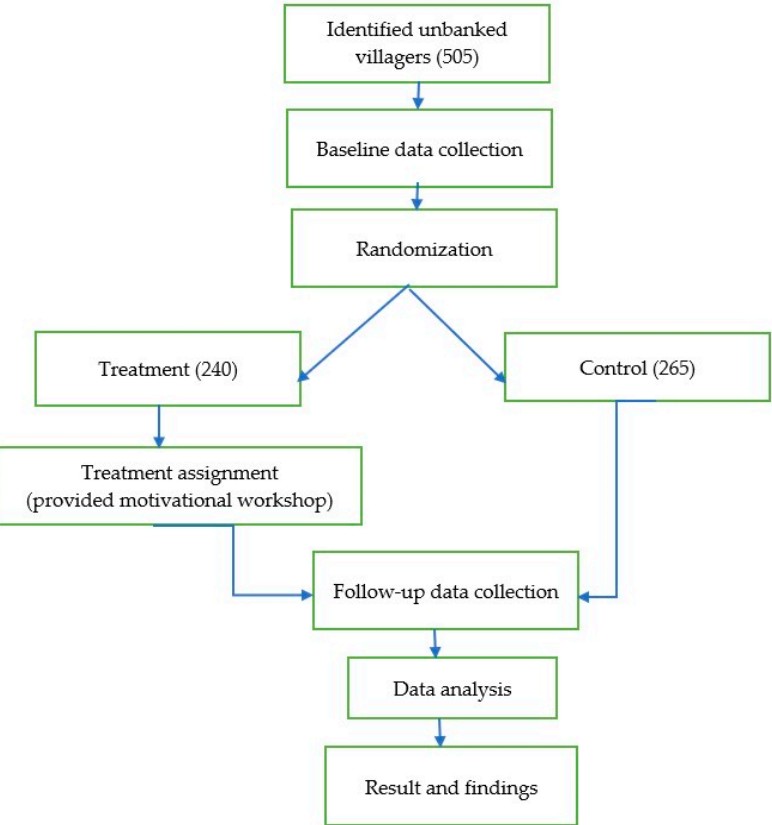

**Figure 2.** Research Design. Figure is based on authors research conduct method.

*3.3. Treatment Assignment*

We send messages to the treatment group to attend the motivational workshop. Before the treatment assignment, the union VDP leaders communicated with all the treatment group members so that they were well-informed about joining the workshop. The treatment group members were invited to a school one time for the treatment assignment. The session was successful since all the members attended the session. The attendance rate is cent percent due to the union leader's communication and treatment group members' acknowledgment of the session.

The local NGO manager and agent bank representative were invited to conduct the workshop. They are well-informed about the national financial inclusion strategy before they conduct the workshop. The synopsis of the workshop is shown in Table 2.

**Table 2.** Intervention session synopsis.

| Intervention Time | One Hour |
|---|---|
| Workshop mode | Oral, interactive discussion, question, and answer session. |
| Discussion issue | Institutional and government offers for opening accounts, the process of opening accounts, benefits and advantages of savings accounts |
| Intervention provider | Local NGO manager, and agent bank employee. |
| Treatment Frequency | Once only for the treatment group |

They conducted a one-hour workshop. The workshop was highly interactive followed by questions and answers during and after the session. The local NGO manager and agent bank representative explained the institutional offer, benefit, process, and required documents for opening a savings account such as an application form, NID card, and passport-size photograph. In conclusion, they emphasize that opening a savings account depends on their willingness. They could apply now or later to open a savings account.

However, no restriction was imposed on them in choosing financial institutions for opening savings accounts.

### 3.4. Data Analysis Approach

The main focus of this study is to examine the impact of motivational workshops on opening savings accounts among unbanked members. Since we randomly assigned our treatment, the mean difference in outcome between the treatment and the control is the unbiased estimation of the average treatment effect (ATE). Theoretically,

$$\text{ATE} = \text{E}[Y_1 | \text{T} = 1] - \text{E}[Y_0 | \text{T} = 0] \tag{1}$$

This study applies the following equation to estimate the impact of motivational workshops on opening savings accounts:

$$Y_i = a_i + \beta_1 T_i + u_i \tag{2}$$

Here $Y_i$ denotes opening a savings account, 1 if the respondent opens a savings account and 0 otherwise. $a_i$, the intercept, represents the constant value of the dependent variable ($Y_i$) when all other independent variables are zero. $T_i$ denotes the treatment variable; attending a motivational workshop 1 if the respondent is in the treatment group and 0 if the respondent is in the control group. $\beta_1$, the coefficient will measure the average treatment effect. The error term will be captured by $u_i$. Moreover, the following equation is used to measure the impact of a motivational workshop on opening a savings account with covariates:

$$Y_i = a_i + \beta_1 T_i + \beta_2 D_i + u_i \tag{3}$$

Here $D_i$ denotes the pre-treatment covariates. The covariates include age, gender, education, total income, total expenditure, agricultural land size, family size, distance from financial institutions, and mobile phone and internet connection possessions.

We calculate the estimation using the OLS regression since the direction of the effect of treatment on outcome is always the same for the logit and probit regression. However, we also perform probit and logit regression for robustness checking (See Appendix A: Table A1).

### 3.5. Summary Statistics

Table 3 demonstrates summary statistics and balance checks based on respondents' demographic and household data at baseline.

The average age of the respondent is more than 33 years. From the demographic dividend's perspective, the respondent belongs to the active age group. Their educational attainment is more than nine years on average, indicating that all the respondents have at least primary education in the Bangladesh context. In terms of income, the summary table shows the average income is 12,592 BDT. The average monthly household income is 26,163 BDT in rural areas (HIES 2023) which is much higher than the respondent's monthly average income. Furthermore, the monthly average expenditure in rural areas is 26,842 BDT (HIES 2023) whereas the respondent's average expenditure per month is 10,242 BDT. The higher difference in income and expenditure indicates the respondents' economic status. Regarding gender, the treatment group has 76% male and 24% female. The control group consists of male 72% and female 28%. In terms of possession of a mobile, almost 97% in the control group and 99% in the treatment group have mobile phones. However, compared to the possession of mobile, the rate of internet users is much lower in both groups; 38% in the control group and 44% in the treatment group use internet.

**Table 3.** Summary statistics and balance check.

| Variables | Total Sample (505) | | Treatment (240) | | Control (265) | | Diff (C-T) and Std Err |
|---|---|---|---|---|---|---|---|
| | Mean | Std Dev | Mean | Std Dev | Mean | Std Dev | |
| Age (year) | 33.71 | 8.34 | 34.12 | 8.15 | 33.34 | 8.51 | −0.78 [0.74] |
| Gender (1 = male, 0 = female) | 0.74 | 0.44 | 0.76 | 0.43 | 0.72 | 0.45 | −0.04 [0.04] |
| Education (year) | 9.17 | 1.6 | 9.25 | 1.63 | 9.1 | 1.57 | −0.16 [0.14] |
| Family size (number) | 4 | 1.16 | 4.06 | 1.2 | 3.95 | 1.13 | −0.11 [0.10] |
| Agricultural land size (decimals) | 28.42 | 23.53 | 29.81 | 22.98 | 27.17 | 24 | −2.64 [2.10] |
| Total income (BDT) | 12,592.08 | 3474.15 | 12,594.58 | 3526.65 | 12,589.81 | 3432.6 | −4.77 [309.88] |
| Total expenditure (BDT) | 10,242.97 | 3266.02 | 10,067.5 | 3299.87 | 10,401.89 | 3233.08 | 334.39 [290.94] |
| Food expenditure (BDT) | 5580.20 | 1461.48 | 5360.42 | 1321.81 | 5779.25 | 1553.14 | 418.83 *** [129.01] |
| Bank distance (meters) | 4509.21 | 2307.55 | 4403.13 | 2336.68 | 4605.28 | 2281 | 202.16 [205.63] |
| MFIs distance (meters) | 1668.32 | 1128.57 | 1661.58 | 1126.19 | 1674.42 | 1132.82 | 12.83 [100.66] |
| MFS distance (meters) | 588.19 | 510.68 | 568.65 | 534.9 | 605.89 | 488.06 | 37.24 [45.52] |
| Mobile user (1 = user, 0 = otherwise) | 0.98 | 0.12 | 0.98 | 0.15 | 0.99 | 0.09 | −0.01 [0.01] |
| Mobile type (1 = smartphone, 0 = otherwise) | 0.41 | 0.49 | 0.39 | 0.49 | 0.44 | 0.5 | −0.05 [0.04] |
| Internet user (1 = user, 0 = otherwise) | 0.41 | 0.49 | 0.38 | 0.49 | 0.44 | 0.5 | −0.06 [0.04] |

Source: Authors own calculation based on field survey. Note: Standard errors are in brackets; level of significance: *** $p < 0.01$.

*3.6. Balance Check*

We conducted a t-test to distinguish the demographic characteristics between the treatment and the control groups (Table 3). Statistically, no significant differences are observed in the balance check after randomly assigning the samples to treatment and control groups. The average age difference between the treatment (34.12) and the control (33.34) is statistically non-significant. However, the higher age of the treatment group indicates that the control group is a little bit younger than their counterpart. There is no significant difference between educational attainment, family size, and agricultural land ownership between the treatment and the control group. Moreover, no observable and statistically significant differences are found in total income, total expenditure, distance from banks, micro-financial institutions, and mobile financial service points from the respondents' houses.

Although all the demographic and household variables are balanced, we observe an imbalance in food expenditure between the control and the treatment groups at a 1% level of significance. The imbalanced variable "food expenditure" is the segment of total expenditure. We observe total expenditure in both groups is balanced, so we can claim that our randomization is almost successful.

## 4. Results

*4.1. Account Opening Take-Up Rate*

Table 4 represents account opening frequency and percentage in different financial institutions (take-up rate) among the treatment group members. Total savings account opening (take-up rate) is 38.75% among the treatment groups. The highest take-up rate is observed through mobile financial services 18.33% which is 44 in number. The second highest rate is found for MFI accounts 26 in number (10.83%) and the bank account opening rate is 25 (10.42%).

**Table 4.** Account opening take-up rate.

| Outcome Variables | Frequency | Percentage |
|---|---|---|
| Total account opening | 93 | 38.75 |
| Account opening in Bank | 25 | 10.42 |
| Account opening in MFIs | 26 | 10.83 |
| Account opening in MFS | 44 | 18.33 |

Note: the summation of account opening in different financial institutions differs from total account opening since some respondents initiate more than one account.

### 4.2. Average Treatment Effect (ATE)

The average treatment effect on opening savings accounts in different financial institutions is presented in Table 5.

**Table 5.** Average treatment effect (ATE) estimation.

| Outcome Variables | Treatment (Mean) | Control (Mean) | ATE | ATE | ATE |
|---|---|---|---|---|---|
| | (1) | (2) | (3) | (4) | (5) |
| Total account opening | 0.3875 [0.4882] | 0.0642 [0.2455] | 0.3233 *** (0.0349) | 0.3166 *** (0.0359) | 0.3149 *** (0.0359) |
| Account opening in the bank | 0.1042 [0.3061] | 0.0038 [0.0614] | 0.1004 *** (0.0201) | 0.099 *** (0.0203) | 0.1023 *** (0.0210) |
| Account opening in MFIs (Micro-financial institution) | 0.1083 [0.3115] | 0.0302 [0.1714] | 0.0781 *** (0.0227) | 0.0835 *** (0.0232) | 0.0721 ** (0.0230) |
| Account opening in MFS (Mobile financial service) | 0.1833 [0.3877] | 0.0302 [0.1714] | 0.1531 *** (0.0272) | 0.1437 *** (0.0278) | 0.1491 *** (0.0278) |
| Observation | 240 | 265 | 505 | 505 | 505 |
| Covariate | No | No | No | Yes # | Imbalanced variable ## |

Note: Robust standard errors are in parentheses; standard deviations are in brackets. Level of significance: *** $p < 0.01$, and ** denotes 0.05 level of significance, # Age, gender, education, marital status, total income, total expenditure, family size, agricultural land size, and distance from banks, MFIs, MFS, mobile, and internet users are used as control variables. ## food expenditure is the imbalanced variable.

The ATE estimation with and without covariates are shown in Table 5. Columns number four and five show robustness. The impact of a motivational workshop on opening a savings account is shown in the third column. The fourth column shows ATE estimation with controlling variables. We use household and respondent's demographic variables such as age, gender, education, total income and total expenditure, family size, agricultural land size, distance from financial institutions to respondents' house, possession of mobile and an internet connection, and type of mobile used by respondents as controlling variables. The last column represents results with imbalanced variables (food expenditure). The treatment effect of the motivational workshop is persistent and significant in three cases. Besides, measuring the treatment effect on total account opening, we also calculated the treatment effect based on respondents' preference for financial institutions to open savings accounts (rows 3, 4, 5). In this regard, we also observed a positive and significant impact of the motivational workshop on opening savings accounts. Furthermore, we also check the robustness of our findings by doing probit and logit regression (see Appendix A, Table A1). The logit regression results show that those who received treatment were 2.22 times more likely to open a savings account. In addition, the probit regression indicates that the intervention increases 1.23 times of opening a savings account.

The treatment induced a 32.33 percent increase in opening savings accounts in total. The increasing rate is consistent in terms of some pre-treatment controlling variables and with imbalanced variables which are 31.66 percent and 31.49 percent, respectively. The intervention improves opening savings accounts in banks by 10.04 percent. It is also

consistent in terms of covariates and with an imbalanced variable which is 9.9 percent and 10.23 percent, respectively. It is at a 1% level significant in all three conditions. We find similarities in opening savings accounts in micro-financial institutions (MFIs) after the intervention. The third row of the table represents the result. It increases MFIs accounts by 7.81 percent. However, the increasing rate is 8.35 percent (1% level significant) and 7.21 percent (5% level significant), respectively, with covariates and imbalanced variables. The fourth row represents the highest increasing rate for mobile financial services among the respondents, which is 15.31 percent significant at the 1% level. Similarly, with covariates and imbalanced variables, the impact of a motivational workshop on opening savings accounts is 14.37 percent and 14.91 percent and both are at a 1% level of significance.

### 4.3. Conditional Average Treatment Effect on Opening Savings Accounts

Policymakers are interested in adopting policies by targeting the population. Therefore, sub-sample analysis is necessary to identify the most appropriate groups where the treatment impact is the most effective. For this, we do a sub-sample analysis to understand the treatment effect on a particular group. In this regard, we segregated the samples based on some observable characteristics. We are inspired by the studies (Cole et al. 2011; Blanco et al. 2020) to select the variables for creating several subgroups. Finally, we estimate the conditional average treatment effect (CATE) concerning educational attainment, age, total income, and gender.

Table 6 represents the sub-sample analysis. We use the average value of educational year, age, total income, and gender for separating the total samples into different groups. We observe that the intervention "motivational workshop" has a persistent and significant impact on different subgroups to open savings accounts. We observe that our CATE estimation is higher than the ATE estimation (32.33%) on total account opening (see Table 5). The CATE estimation is higher than the ATE estimation of total account opening for the secondary education groups (n = 123), age group equal and below 33 years (n = 260), and income groups equal and above 12,592 BDT (n = 233). Our CATE estimation analysis is almost equal for both males and females. The highest CATE calculation is noticed at 40.23% for the secondary education groups.

**Table 6.** Conditional Average Treatment Effect on Opening Savings Account.

| Category | Sub Sample | CATE |
|---|---|---|
| | Primary education (n = 296) | 0.2890 ***<br>(0.0455) |
| Education | Secondary education (n = 123) | 0.4023 ***<br>(0.0690) |
| | More than higher secondary education (n = 86) | 0.3139 ***<br>(0.0883) |
| Age | Age $\leq$ 33 (n = 260) | 0.3658 ***<br>(0.0510) |
| | Age > 33 (n = 245) | 0.2857 ***<br>(0.0478) |
| Total Income | Income $\geq$ 12592 (n = 233) | 0.3415 ***<br>(0.0490) |
| | Income < 12592 (n = 272) | 0.3097 ***<br>(0.0495) |
| | Male (n = 374) | 0.3252 ***<br>(0.0402) |
| Gender | Female (n = 131) | 0.3184 ***<br>(0.0713) |

Note: Robust standard errors are in parentheses; level of significance: *** $p < 0.01$. n = number of respondents in the group.

Table 7 represents the descriptive statistics of outcome variables for the whole sample, treatment, and control group. The impact of the motivational workshop on opening a savings account for the whole sample is 21.78 percent, institutionally 5.15 percent for bank accounts, 6.73 percent for micro-financial accounts, and 10.30 percent for mobile financial services accounts. The intervention effect has 13.55 times, 2.24 times, and 3.42 times higher magnitude for the total sample in terms of opening savings accounts in banks, MFIs, and MFS than the control group.

**Table 7.** Descriptive statistics for outcome variables.

| Outcome Variable | Total Sample n = 505 | | Treatment n = 240 | | Control n = 265 | | Magnitude |
|---|---|---|---|---|---|---|---|
| | Mean | Std Dev | Mean | Std Dev | Mean | Std Dev | |
| Total account opening | 0.2178 | 0.4132 | 0.3875 | 0.4882 | 0.0642 | 0.2455 | 3.39 |
| Account opening in Bank | 0.0515 | 0.2212 | 0.1042 | 0.3061 | 0.0038 | 0.0614 | 13.55 |
| Account opening in MFIs | 0.0673 | 0.2508 | 0.1083 | 0.3115 | 0.0301 | 0.1714 | 2.24 |
| Account opening in MFS | 0.1030 | 0.3042 | 0.1833 | 0.3877 | 0.0301 | 0.1714 | 3.42 |

Note: Magnitude $= \frac{\text{Total sample mean}}{\text{Controlled group mean}}$.

## 5. Discussion

Motivational workshop for the unbanked is successful in achieving the aim of increasing opening savings accounts. The intervention enhances opening accounts by 32.33% percent among the unbanked villagers. The intervention "motivational workshop" is statistically significant in terms of whether we control some pre-treatment variables or not. Moreover, our findings are statistically persistent and consistent even in terms of imbalanced variables. Account opening take-up rate differs in terms of choosing different financial institutions since the respondents were free to choose financial institutions for opening accounts. We observe the bank accounts' opening rate of 10.04% followed by micro financial institution accounts at 7.81%. However, the highest take-up rate is observed for mobile financial services (15.31%) for various reasons such as easy account opening options, availability, less documentation process, 24/7 service even from a feature phone, zero account opening fee, and physical proximity of mobile financial services. In this regard, Khatun et al. (2021) also explore that cash in and out facilities, person-to-person pay (P2P), government-to-person pay (G2P), utility payment facilities, receiving government subsidies, and salary payment accelerate the popularity of mobile financial services in Bangladesh.

The findings of this study are consistent with (Karakara and Osabuohien 2019; Hoy et al. 2022; Prina 2015) in terms of the positive direction of opening a savings account, although their intervention is different from this study. Some experimental studies show the highest take-up rate: 84% among the women slum dwellers in Nepal (Prina 2015), and a 70% take-up rate among the unbanked villagers in Papua New Guinea (Hoy et al. 2022). In comparison to some previous studies, our take-up rate is modest, and our intervention demonstrates a strong and positive impact considering intervention cost and impactful response within a short time. Firstly, we observed the impact of a one-hour-long motivational workshop only on opening savings accounts. Secondly, we rely on a motivational workshop without any monetary subsidy and assistance to ease the documentation process. Finally, we provided a short period for collecting the follow-up data on opening a savings account just after 45 days of assigning intervention. Dupas et al. (2018) conducted their study and found a take-up rate of 69% in Malawi, 54% in Uganda, and 17% in Chile after three rounds (6 months, 12 months, and 18 months) of follow-ups. Whereas Chin et al. (2010) observed that 38% were more likely to open bank accounts among the respondents after 5 months. The modest outcome of this study in comparison with the existing literature can be attributed to different settings, intervention modes, and contextual variations. Despite having a one-hour intervention with existing conditions

of opening saving accounts, we find that motivational workshops can enhance financial access and bring the unbanked under the umbrella of financial inclusion. The findings have some policy implications since low-cost interventions are viable for policymakers in developing countries Vu et al. (2020). The intervention is cost-effective and easily replicable in different contexts because of its simplicity and short but interactive motivation giving the necessary information for opening a savings account.

We have a keen interest in monitoring whether the unbanked women, the poor, and villagers open an account or not. Our sub-sample analysis vibrantly expresses that motivational workshop positively inspires the noted groups to open savings accounts. The CATE analysis is the highest at 40.23 percent for those who have secondary education among the unbanked. The CATE for males (32.52 percent) and females (31.84 percent) is almost like the ATE (32.33 percent) on total account opening. From our sub-sample analysis, it is observed that the CATE estimation is higher than the ATE estimation for several categories, such as 1.24 times higher for those who have secondary education, and 1.05 times higher for the income group who have income more than 12592 BDT. Furthermore, CATE is 1.06 times higher for the younger age group less than and equal to 33 years (see Appendix A: Table A2). Considering the higher magnitudes of CATE, policymakers can adopt appropriate policies for the respective groups from this finding.

Another interpretation of the significant increase in opening savings accounts is due to well understanding of the content of the workshop within a short time. Because of their good understanding, the respondents have chosen their financial institutions based on their preferences and needs. The respondents have no shortage of documents for opening accounts as the necessary documents for opening an account are similar to those of buying a mobile phone. The possession of necessary documents accelerates the account opening process. Moreover, all the respondents are more than 18 years old. Thus, they are psychologically mature enough to understand and make positive decisions. Furthermore, the essence of the workshop is to motivate the unbanked to open savings accounts. Motivation is the key element for expected behavioral change as it initiates, and guides towards a goal-oriented change (Rodgers and Loitz 2008). Our motivational workshop demonstrates a significant increase in opening savings accounts since any expected outcome can be possible if the weightage of motivation is higher than other issues (Willmott et al. 2021).

## 6. Conclusions

Despite having a good number of options for opening accounts in financial institutions, account ownership is low in Bangladesh. Only 53% of adults in Bangladesh have bank accounts. Still, many people are financially excluded which provokes the financially excluded to adopt risky, inconvenient, and informal mechanisms for saving, sending, and receiving money. Based on our study and data analysis, we have evidence that motivational workshop has a positive impact on opening savings accounts in financial institutions among unbanked people. Motivation triggers changing a positive mindset. The motivational workshop is successful since it has enhanced the take-up rate among the unbanked villagers by 38.75 percent. The effectiveness of the intervention is much higher in opening saving accounts through mobile financial services by 15.31% followed by bank accounts at 10.04% and micro-financial accounts at 7.81%. Our CATE analysis is also statistically significant for different target groups, especially for female and secondary education groups. Our CATE analysis indicates that unbanked people regarding the classification of age, gender, and education, all have a statistically significant impact on opening savings accounts.

The findings of this study have significant policy implications for adopting financial inclusion strategies for developing countries. Effective policy on financial access can move unbanked people from financial exclusion to financial inclusion. We can suggest policymakers adopt motivational workshops for financially excluded people. This finding can help implement the National Financial Inclusion Strategy of the Bangladesh (NFIS-B) government. The NFIS-B has a target to bring the typically unbanked and marginalized people in

terms of little income and educational attainment among tea laborers, physically impaired people, third gender, floating communities, slum dwellers, people in geographically remote areas such as forest, coastal, *haor* (vast marshy wetland), *char land* (sandy island beside the river) under financial inclusion. Furthermore, our intervention is pertinent as it is easily replicable and less expensive than the studies provided money subsidy. The Bangladesh government targets to ensure universal financial inclusion by 2026. The NFIS-B proposed financial literacy and an annual program for achieving the goal. In line with the strategies of the NFIS-B, the stakeholders may adopt our intervention "motivational workshop" for the unbanked population as an annual program. In addition, the Bangladesh Ansar and VDP organizations have much scope for adopting motivational workshops in their village-based training program to achieve the goal of the NFIS-B.

This study has some limitations. Spillover is one of them since we did our randomization at the individual level. Moreover, our study area is small. Thus, there is a higher probability of exchanging intervention messages from the treatment group to the control group. As a result, our estimation may be underestimated.

Despite that, our findings are still positive and statistically significant. Thus, we encourage further research firstly, to examine the long-term effect of motivational workshops among unbanked people, secondly, to observe the saving volume and active account usage tendency after opening accounts, and thirdly, to explore the impact of motivational workshops in geographically challenged areas.

**Author Contributions:** Conceptualization, M.M.M. and K.L.M.; methodology, M.M.M.; software, M.M.M.; validation, M.M.M. and K.L.M.; formal analysis, investigation, resources, data curation, M.M.M.; writing—original draft preparation, M.M.M.; writing—review and editing, M.M.M. and K.L.M.; visualization, M.M.M. and K.L.M.; supervision, K.L.M. All authors have read and agreed to the published version of the manuscript.

**Funding:** There was no external funding for this research.

**Institutional Review Board Statement:** The study complies with the guideline and prior approval (application no: HR-LPES-000463) from the Ethics Committee of the Graduate School for International Development and Cooperation (IDEC), Hiroshima University, Japan, on 31 August 2022.

**Informed Consent Statement:** All the unbaked respondents provided their informed consent.

**Data Availability Statement:** The data presented in this study are available on request from the corresponding author.

**Acknowledgments:** The author would like to thank the project for the Human Resource Development Scholarship (JDS) organized by the Japan International Cooperation Center (JICE), which facilitated the master's study of M.M.M. at Hiroshima University, Japan.

**Conflicts of Interest:** The authors declare no conflict of interest.

## Appendix A

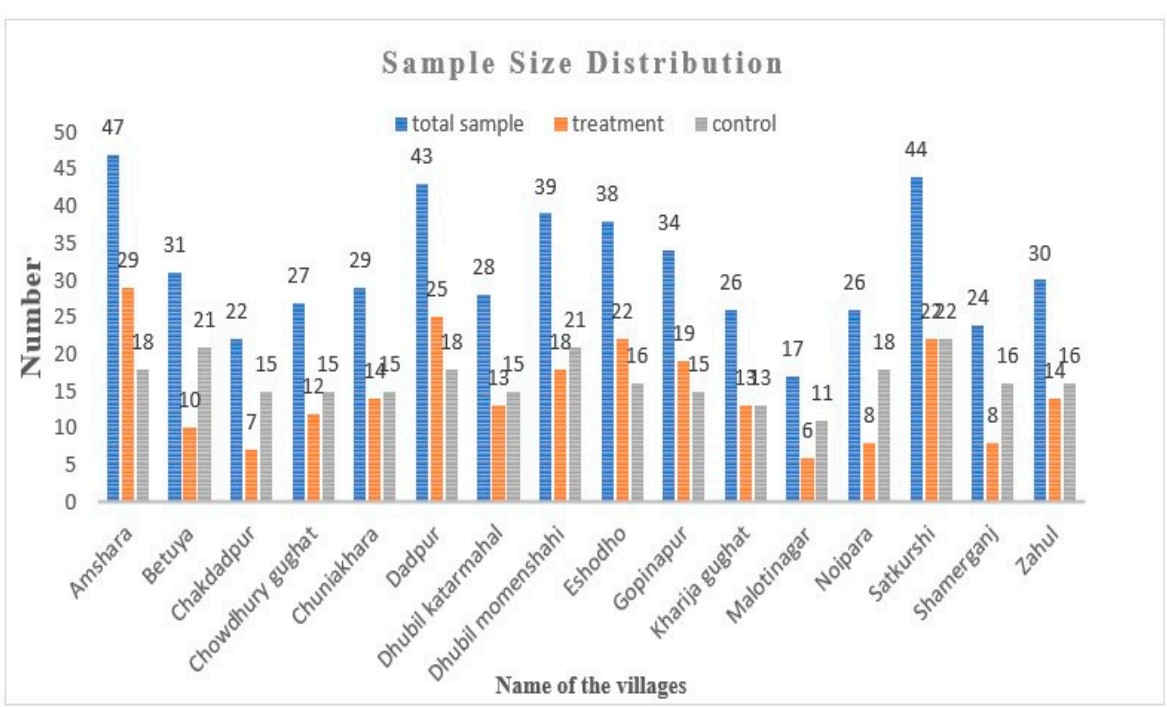

**Figure A1.** Village-wise sample size distribution.

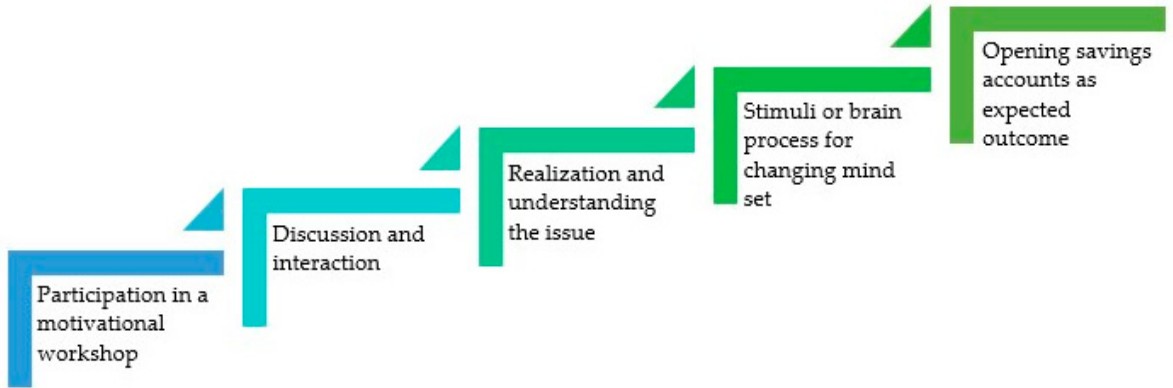

**Figure A2.** Conceptual framework process of motivational workshop.

**Table A1.** Logit and probit estimation for checking the robustness.

| Outcome | Coefficient and Std Err | Coefficient and Std Err |
|---|---|---|
| Total account opening | 2.22 *** (0.2838) | 1.23 *** (0.1455) |
| Bank account opening | 3.42 *** (1.025) | 1.41 *** (0.3526) |
| MFIs account opening | 1.36 *** (0.4151) | 0.6426 *** (0.1879) |
| MFS account opening | 1.98 *** (0.3962) | 0.9753 *** (0.1804) |
| Regression | Logit | Probit |
| Observation | 505 | 505 |

Note: Robust standard errors are in parentheses; level of significance: *** $p < 0.01$.

**Table A2.** The magnitude of CATE.

| Sub Sample | CATE | ATE | Magnitudes |
|---|---|---|---|
| Secondary education (n = 123) | 0.4023 | | 1.24 |
| Age $\leq$ 33 (n = 260) | 0.3658 | 0.3233 | 1.13 |
| Income $\geq$ 12,592 (n = 233) | 0.3415 | | 1.05 |

Note: Magnitude $= \frac{CATE}{ATE}$.

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
