# Peer review of "Impact of Motivational Workshop on Financial Inclusion of Rural People in Bangladesh: Evidence from Randomized Controlled Trial"

_ijfs, doi:10.3390/ijfs11040151_

Round 1

Reviewer 1 Report

Comments and Suggestions for Authors

Summary of the Paper:

The authors explore the influence of motivational workshops on the likelihood of unbanked individuals, particularly members of the Village Defense Party (VDP), to initiate savings accounts. Presently, only half of adults in Bangladesh possess financial accounts. This statistic underscores the significant segment of the population that remains financially excluded, relying on precarious and inconvenient methods of money management. Utilizing a randomized controlled trial approach, the study discerns a pronounced positive impact of these workshops. The research underscores that the motivational workshop can markedly enhance the rate of account openings within this demographic, especially promoting mobile financial services. These insights carry profound implications for strategies aiming at financial inclusion in developing nations and resonate with the objectives of Bangladesh’s National Financial Inclusion Strategy.

Referee Comments:

1. Overview:

The manuscript delivers a meticulously conducted study, elucidating the pronounced effects of motivational workshops on the financial decisions of the unbanked populace. It augments our comprehension of how such interventions can propel financial inclusivity in emerging economies. The ensuing suggestions aim to hone and elucidate the narrative and discourse, ensuring the study’s pivotal contributions and overarching implications are lucidly conveyed.

2. Structure and Clarity:

The research’s impetus and central theme emerge relatively late, particularly in the denouement of the Introduction. This structuring could potentially obscure readers’ grasp and inadvertently eclipse the study’s novel contributions and its delineation from extant literature (notably, some of these distinctions aren’t broached until the Discussion section). I advocate for a prompt introduction to the study’s inspiration, principal objectives, and research queries. This should be succeeded by a literature survey and foundational context, culminating in a précis of the pivotal findings and their ramifications, offering a more reader-friendly architecture.

3. Originality:

While the manuscript furnishes salient outcomes, it mirrors several facets of established literature. Given the extensive citations to prior works, it becomes imperative for the authors to spotlight the singular facets of this research. Elucidating how their work either diverges from or supplements existing studies will accentuate its innovative essence.

4. Treatment Clarity:

The study’s methodology capitalizes on a “motivational workshop” as its treatment modality. While descriptions are proffered, integrating a demonstrative example (potentially within an appendix) that elucidates the workshop’s content and format would empower readers with a deeper insight into the treatment’s core. 

5. Alternative Financial Systems:

The manuscript alludes to a significant fraction of the populace devoid of formal financial accounts but refrains from probing potential alternative (and perhaps informal) financial mechanisms they might resort to. A discourse on the dependence of these individuals on non-formal financial conduits, such as local savings consortiums or other clandestine financial channels, becomes paramount. Discrepancies in the engagement levels with these services among the experimental cohort could sway their inclination to inaugurate formal savings accounts. A discourse on any prospective biases this may induce is warranted.

Comments on the Quality of English Language

Typo correction.

Author Response

Thank you very much for good review comments. our reply is attached herealong.

Reviewer 2 Report

Comments and Suggestions for Authors

Overall, this study design for the topic is generally acceptable. However, (1) it would be beneficial to highlight the significant contribution of this study compared to previous ones that have already been conducted on the same issue and design.

(2) Additionally, the literature review in this study needs to be substantially refined as it currently appears fragmented, loosely connected, and incoherent.

(3) Third, the format used in the footnotes reveals a lack of professionalism and should be addressed.

(4) It would be helpful to clarify what "BDT" stands for, specifically that it refers to Bangladesh Taka, for the benefit of readers.

(5) Table 4 could be merged with Table 3, and while adding an additional column to Table 3 is acceptable, creating a new table seems unnecessary.

(6) Similarly, Tables 6 and 7 can be improved by adding just two additional columns to Table 6 instead of creating a separate table.

(7) In Table 8, it would be helpful to explain the differences between the two results columns. Consider adding a column title or a table note to clarify these differences, as it was difficult to understand from the current description.

(8) Lastly, please review the References section for any formatting errors that may need to be checked thoroghly.

Author Response

Thank you very much for the good comments. Our reply is attached herewith.
